# Multi-Stage Pre-training Enhanced by ChatGPT for Multi-Scenario Multi-Domain Dialogue Summarization

**Weixiao Zhou**[†]    **Gengyao Li**[‡§]    **Xianfu Cheng**[†]    **Xinnian Liang**[†]
**Junnan Zhu**[‡§]    **Feifei Zhai**[¶]    **Zhoujun Li**[†*]

[†]State Key Lab of Software Development Environment, Beihang University
[‡]State Key Lab of Multimodal Artificial Intelligence Systems, Institute of Automation, CAS
[§]School of Artificial Intelligence, University of Chinese Academy of Sciences
[¶]Fanyu AI Research, Zhongke Fanyu Technology Co., Ltd
{wxzhou,buaacxf,xnliang,lizj}@buaa.edu.cn    junnan.zhu@nlpr.ia.ac.cn

## Abstract

Dialogue summarization involves a wide range of scenarios and domains. However, existing methods generally only apply to specific scenarios or domains. In this study, we propose a new pre-trained model specifically designed for multi-scenario multi-domain dialogue summarization. It adopts a multi-stage pre-training strategy to reduce the gap between the pre-training objective and fine-tuning objective. Specifically, we first conduct domain-aware pre-training using large-scale multi-scenario multi-domain dialogue data to enhance the adaptability of our pre-trained model. Then, we conduct task-oriented pre-training using large-scale multi-scenario multi-domain "*dialogue-summary*" parallel data annotated by ChatGPT to enhance the dialogue summarization ability of our pre-trained model. Experimental results on three dialogue summarization datasets from different scenarios and domains indicate that our pre-trained model significantly outperforms previous state-of-the-art models in full fine-tuning, zero-shot, and few-shot settings[1].

## 1 Introduction

Dialogue summarization is the task of generating a summary from a dialogue (Xu et al., 2022). Specifically, open-domain dialogue summarization involves various scenarios (e.g., Online-Chat (Gliwa et al., 2019) and Daily-Life (Chen et al., 2021)), while customer service dialogue summarization involves different domains (e.g., Tweet (Feigenblat et al., 2021) and E-commerce (Lin et al., 2022)).

Recently, general-purpose pre-trained models have achieved significant success in dialogue summarization tasks (Lewis et al., 2020; Bao et al., 2020; Beltagy et al., 2020). Furthermore, several task-specific pre-trained models (Zhang et al., 2020a; Zhong et al., 2022) have further improved

dialogue summarization. Existing dialogue summarization pre-trained model (Zhong et al., 2022) achieves good performance on long dialogue summarization. However, it still has the following limitations: (1) It is only pre-trained on dialogue corpora that include two domains (i.e., Interview and TV show), making it difficult to apply to dialogue summarization in a wide range of scenarios and domains. (2) It utilizes a window-based denoising task as the pre-training objective, which presents a significant gap with the fine-tuning objective. Simultaneously, existing state-of-the-art (SOTA) models generally improve dialogue summarization by modeling dialogue interactions (Lin et al., 2022; Tang et al., 2022), incorporating extra information (e.g., topics and roles) (Wang et al., 2022c; Kim et al., 2022), and rewriting dialogues (Xu et al., 2022; Fang et al., 2022). Although these methods have some effect, they still have limited applicability to downstream datasets in different scenarios and domains and are often difficult to apply within the current pre-training paradigm due to complex model architectures.

To address the limitations of previous works, in this study, our goal is to propose a task-specific pre-trained model for dialogue summarization, which has extremely small gap between the pre-training objective and the fine-tuning objective, enabling it to excellently adapt to downstream datasets from a wide range of scenarios and domains in full fine-tuning, few-shot, and zero-shot settings.

Motivated by the above goal, we consider three key components in the implementation of our pre-trained model: model architecture, pre-training corpus, and pre-training strategy. For **model architecture**, our pre-trained model is based on the standard Transformer (Vaswani et al., 2017) encoder-decoder architecture and is initialized with BART (Lewis et al., 2020). To capture the underlying role interactions during the dialogue process, we incorporate additional speaker embed-

---

[1]Our corpus and enhanced pre-trained models can be found at https://github.com/zhouweixiao/MP4
*Corresponding Author

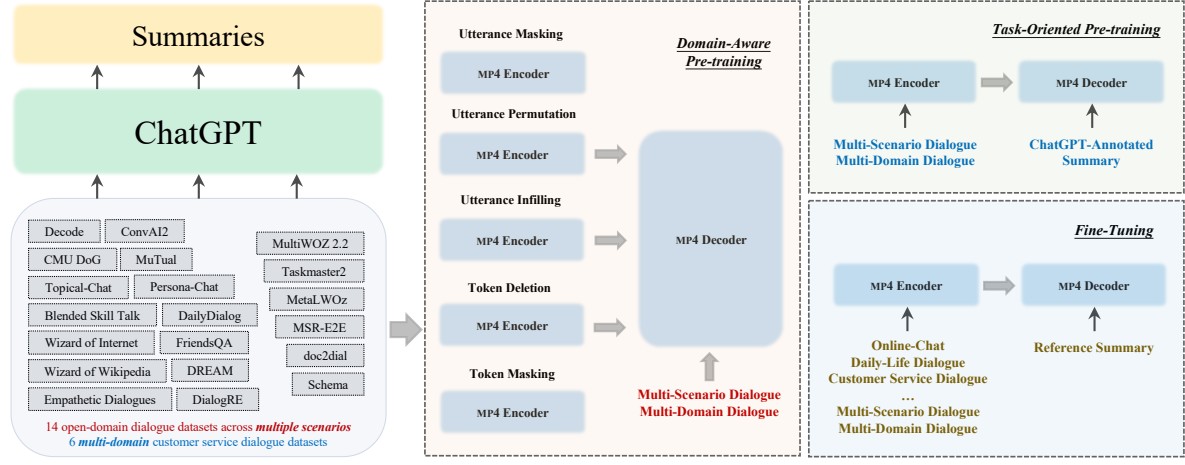

(a) The construction process of Lᴄᴍ³Dꜱ    (b) Multi-stage pre-training and fine-tuning for Mᴘ4

Figure 1: Our pre-training corpus Lᴄᴍ³Dꜱ and pre-trained model Mᴘ4.

dings (Gu et al., 2020, 2021) into token representations. For **pre-training corpus**, we collect 14 open-domain dialogue datasets across multiple scenarios and 6 multi-domain customer service dialogue datasets. Furthermore, due to the development of Large Language Models (LLMs) (Zeng et al., 2022; Thoppilan et al., 2022; Scao et al., 2022) and their excellent generative ability, obtaining high-quality "*dialogue-summary*" parallel pre-training data has become possible. Therefore, we utilize ChatGPT (Ouyang et al., 2022) to annotate the collected multi-scenario multi-domain dialogues and obtain corresponding summaries. We refer to our pre-training corpus as Lᴄᴍ³Dꜱ (**L**arge-scale **C**hatGPT-annotated **M**ulti-scenario **M**ulti-domain **M**ulti-turn **D**ialogue **S**ummarization) (see Figure 1 (a)). For **pre-training strategy**, we conduct multi-stage pre-training to reduce the gap between the pre-training objective and the fine-tuning objective. Specifically, we first conduct domain-aware pre-training using the dialogue data from Lᴄᴍ³Dꜱ to enhance the adaptability of pre-trained model to dialogues in multiple scenarios and domains. Then, we utilize the "*dialogue-summary*" parallel data from Lᴄᴍ³Dꜱ for task-oriented pre-training to enhance the ability of pre-trained model to summarize multi-scenario multi-domain dialogues. We refer to our pre-trained model as Mᴘ4 (**M**ulti-stage **P**re-trained **M**odel for **M**ulti-scenario **M**ulti-domain Dialogue Summarization) (see Figure 1 (b)).

We evaluate our pre-trained model on open-domain dialogue summarization datasets from two scenarios (i.e., Online-Chat (Gliwa et al., 2019) and Daily-Life (Chen et al., 2021)), as well as a cus-tomer service dialogue summarization dataset from a specific domain (i.e., Tweet (Feigenblat et al., 2021)). The experimental results indicate that Mᴘ4 significantly outperforms previous SOTA models in full fine-tuning, zero-shot, and few-shot settings, demonstrating remarkable performance improvements.

Our contributions are summarized as follows:

- We construct Lᴄᴍ³Dꜱ, which includes a large-scale collection of multi-scenario multi-domain dialogues and their corresponding summaries annotated by ChatGPT.

- We propose Mᴘ4, a multi-stage pre-trained model for multi-scenario multi-domain dialogue summarization.

- Our pre-trained model achieves new state-of-the-art performance on three dialogue summarization datasets from different scenarios and domains in full fine-tuning, zero-shot, and few-shot settings.

## 2 Lᴄᴍ³Dꜱ Corpus

### 2.1 Dialogue Preparation

**Dialogue Collection.** We collect 20 high-quality human-to-human multi-turn dialogue datasets to construct the dialogue part of Lᴄᴍ³Dꜱ, including 14 open-domain dialogue datasets across multiple scenarios (Rashkin et al., 2019; Nie et al., 2021; Dinan et al., 2019; Komeili et al., 2022; Gopalakrishnan et al., 2019; Zhang et al., 2018; Smith et al., 2020; Dinan et al., 2018; Li et al., 2017; Cui et al., 2020; Zhou et al., 2018; Yu et al.,

| Dataset | Sce./Dom. | #Dialogue | #Tokens/dial. | #Tokens/summ. | #Comp. | #Cov. | #Dens. | % of novel $n$-grams | | | % of redundant $n$-grams | | |
|---|---|---|---|---|---|---|---|---|---|---|---|---|---|
| | | | | | | | | unigram | bigram | trigram | unigram | bigram | trigram |
| SAMSum | ODDS-Online | 16,368 | 145.1 | 25.3 | 5.99 | 0.71 | **1.51** | 34.24 | **78.98** | **89.54** | 11.78 | 1.04 | **0.20** |
| DIALOGSUM | ODDS-Daily | 13,460 | 208.9 | 34.5 | 6.48 | **0.81** | 2.20 | 26.80 | 63.95 | 81.70 | 19.53 | 6.31 | 2.79 |
| TWEETSUMM | CSDS-Tweet | 1,087 | 328.8 | 39.5 | 7.54 | 0.75 | 2.39 | 32.83 | 73.31 | 83.73 | 16.99 | 1.32 | 0.24 |
| LCM³DS | **Multiple** | **206,768** | 211.3 | 51.9 | **4.03** | 0.73 | 1.88 | **34.54** | 74.19 | 86.17 | 21.87 | 3.00 | 0.66 |

Table 1: Comparison between LCM³DS and existing dialogue summarization datasets. *Sce.* denotes scenario of Open-Domain Dialogue Summarization (ODDS). *Dom.* denotes domain of Customer Service Dialogue Summarization (CSDS). # represents the average value. *dial.* represents dialogue. *Summ.* represents reference summary or ChatGPT-annotated summary. *Comp.* indicates compression ratio. *Cov.* indicates coverage. *Dens.* indicates density.

2020; Yang and Choi, 2019; Sun et al., 2019) and 6 multi-domain customer service dialogue datasets (Lee et al., 2019; Rastogi et al., 2020; Byrne et al., 2019; Zang et al., 2020; Li et al., 2018; Feng et al., 2020). In total, there are 105,426 open-domain dialogues containing over 1M utterances and 101,342 customer service dialogues containing over 1.4M utterances. We provide the details of data statistics for all dialogue datasets in Appendix G.

**Dialogue Pre-Processing and Cleaning.** We conduct a series of automated data pre-processing and cleaning to further improve the quality of the dialogues. For pre-processing, we perform the following steps: (1) Normalizing punctuations, special characters, and capitalization in each dialogue. (2) Following previous studies (Dinan et al., 2019; Chen et al., 2021), we preprocess each dialogue into a dual-turn dialogue format by merging consecutive utterances from the same speaker. For cleaning, we perform the following steps: (1) Removing duplicate and highly similar dialogues using the *Jaccard* text similarity algorithm. (2) Deleting highly similar dialogues **between the dialogue datasets and the evaluation datasets** using the same algorithm as in (1), **ensuring that they have no intersection**. (3) Removing dialogues with less than 4 turns or 32 tokens.

**Role Adding.** In order to standardize the different speaker formats of original dialogues across various datasets, we collect a list containing over 4,000 real names. For each dialogue, we randomly selected several real names from the list to assign a role group (e.g., *Danny* and *Alejandra*), where the number and order of real names in each role group corresponds to the speakers in the original dialogue.

## 2.2 Annotation

**Prompt Format.** We follow the previous study of InstructGPT (Ouyang et al., 2022) by inserting

the text "*Tl;dr:*" at the end of each dialogue as a prompt and inputting it into ChatGPT[2] (in zero-shot setting) to obtain annotated summaries. We also investigate the performance of three different prompts for dialogue summarization in zero-shot setting, and the details can be found in Appendix A.

**Role-Replaced Data Augmentation.** In dialogue summarization, there are multiple scenarios and domains involving different roles. To alleviate this problem, we propose a simple yet effective method that can be extended to dialogue summarization involving any role. Specifically, we directly replace the roles in the dialogues and summaries from LCM³DS to obtain an augmented parallel corpus. In this study, we perform replacements for two common types of roles, including named coreference and customer service. The example we provide can be found in Appendix C.

## 2.3 Data Analysis

We empirically compare LCM³DS with existing dialogue summarization datasets (Gliwa et al., 2019; Chen et al., 2021; Feigenblat et al., 2021) based on five metrics (Grusky et al., 2018; Fabbri et al., 2021): *Compression Ratio*, *Coverage*, *Density*, *Novelty*, and *Redundancy* (see Table 1).

Compared to existing dialogue summarization datasets, LCM³DS exhibits lower *Compression Ratio*, moderate *Coverage*, and lower *Density*, indicating that the summaries maintain a high degree of abstraction while covering the important content of the dialogue and retaining more details and information from the dialogue. Additionally, LCM³DS shows higher *Novelty* and *Redundancy*, which is mainly caused by the lower *Compression Ratio*. Furthermore, unlike existing small-scale, human-annotated, single-scenario, single-domain dialogue summarization datasets, LCM³DS is large-scale, ChatGPT-annotated, multi-scenario, and multi-domain.

---

[2]We use *gpt-3.5-turbo-0301*

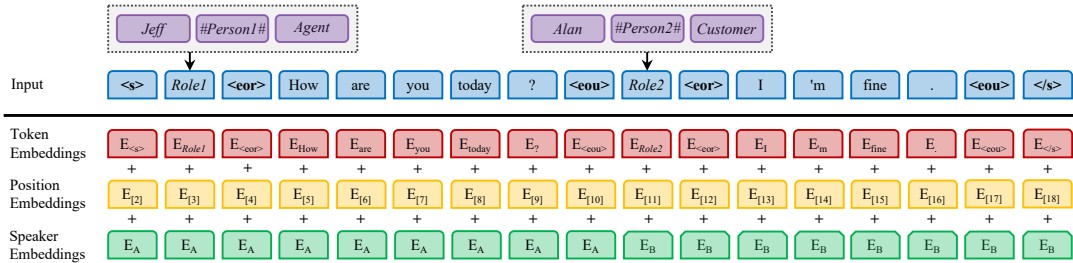

Figure 2: Dialogue modeling of MP4.

## 3 Model

### 3.1 Dialogue Modeling

MP4 is based on the standard Transformer (Vaswani et al., 2017) encoder-decoder architecture. For multi-turn dialogue modeling, the input embedding of each token is the sum of the corresponding token, position, and speaker embeddings. Figure 2 illustrates the dialogue modeling of MP4.

**Input Structure.** Given the dialogue context $D$, we first concatenate all roles $R_i$ and utterances $U_j$ in the dialogue context with two additional special tokens as a separate, consecutive token sequence: $X = \{R_1\texttt{<eor>}U_1\texttt{<eou>}\ldots R_m\texttt{<eor>}U_n\texttt{<eou>}\}$ where the special end-of-role token <eor> and end-of-utterance token <eou> are respectively appended to the end of each role and utterance for multi-turn dialogue separation. Then, we add the start token  and the end token  around the token sequence $X$ as the input for MP4.

**Speaker Embeddings.** To distinguish utterances in the dialogue context and capture underlying role interactions during the dialogue process, we follow previous dialogue modeling studies (Gu et al., 2020, 2021) and add additional speaker embeddings to token representations. This process is performed alternately based on the role transitions and can be extended to dialogues with an unlimited number of roles. The speaker embeddings are combined with the initial token and position embeddings and then fed into the MP4 encoder-decoder framework.

### 3.2 Multi-Stage Pre-training

We conduct multi-stage pre-training to reduce the gap between the pre-training objective and the fine-tuning objective. Domain-aware pre-training aims to enhance the adaptability of MP4 to dialogues in multiple scenarios and domains, while task-oriented pre-training aims to enhance the ability of MP4 to summarize unstructured spoken multi-scenario multi-domain dialogues into structured written-language summaries.

### 3.2.1 Domain-Aware Pre-training

General-purpose pre-trained models (Lewis et al., 2020) are pre-trained on free-form text data with universal pre-training objectives, limiting their ability in specific domains and tasks. Therefore, it is common practice to further train these models with the language modeling objective using text from the target domain to reduce negative impact (Zhang and Zhao, 2021; Whang et al., 2021). In this study, we conduct a domain-aware pre-training stage on MP4 using the dialogue data from LCM³DS. Specifically, we achieve this by modeling a series of **dialogue reconstruction pre-training objectives** inspired by BART (Lewis et al., 2020).

**Token Masking.** For tokens of each utterance in the dialogue, 20% of them are randomly sampled and replaced with a special <mask> token.

**Token Deletion.** 20% of the tokens in the dialogue utterances are randomly sampled and deleted.

**Utterance Infilling.** Several utterance spans are randomly sampled, and each span is replaced with a single <mask> token. The length of each utterance span is drawn from the Poisson Distribution ($\lambda = 3$). 0-length spans correspond to the insertion of <mask> tokens.

**Utterance Permutation.** The order of all utterances in the dialogue turns is randomly shuffled. In contrast to previous studies (Zhong et al., 2022; Wang et al., 2022b), we did not shuffle the order of roles. Therefore, MP4 needs to reconstruct the correct order of utterances and ensure the precise alignment between utterances and roles.

**Utterance Masking.** 20% of the utterances in the dialogue are selected and replaced with a special <uttr-mask> token. We did not perform random

selection but instead followed the method of PE-GASUS (Zhang et al., 2020a) using greedy search to obtain the principal Gap-utterances. During the decoding process, MP4 needs to reconstruct the complete dialogue.

**Multi-Task Learning.** The model is trained with a maximum likelihood objective $\mathcal{L}_\Theta$. Given the training sample $D = (x, y)$, $\mathcal{L}_\Theta$ is defined as

$$\mathcal{L}_\Theta = -\sum_{i=1}^{|y|} \log P_\Theta(y_i|y_{<i}; x) \qquad (1)$$

where $\Theta$ is the model parameters, $x$ is the noisy dialogue, and $y$ is the original dialogue.

During each iteration of the multi-task domain-aware pre-training stage, training samples are randomly selected from different pre-training tasks as mini-batches and used to calculate the cumulative loss and optimize the model parameters $\Theta$.

### 3.2.2 Task-Oriented Pre-training

Several task-specific summarization pre-trained models (Zhang et al., 2020a; Xiao et al., 2022; Zhong et al., 2022) reduce the gap with downstream datasets by modeling the task-oriented pre-training objective. Specifically, They typically select segments (e.g., gap-sentences or window-based utterances) of the original text (e.g., document or dialogue) as optimization targets for the decoder. Although they have some effects, however, there still exists a significant gap between the segments selected through unsupervised methods and abstractive written-language summaries. In this study, we directly utilize the "*dialogue-summary*" parallel data from LCM³DS for task-oriented pre-training stage. The learning objective is similar to Eq. (1), where the training sample $D = (x, y)$, with $x$ representing the original dialogue and $y$ representing the summary annotated by ChatGPT.

## 4 Experiments

### 4.1 Experimental Setup

**Implementation Details.** MP4 is initialized with BART-large[3] (Lewis et al., 2020). which is a denoising sequence-to-sequence pre-trained Transformer (Vaswani et al., 2017) model with 12 layers and 16 attention heads. To facilitate performance comparison, we have implemented four types of MP4 models. MP4 (VANILLA) represents the non-pretrained model, which includes only specialized input structure and speaker embeddings. MP4 (DAP) denotes domain-aware pre-training applied to MP4 (VANILLA). MP4 (TOP) signifies task-oriented pre-training applied to MP4 (VANILLA). MP4 (DAP-TOP) indicates multi-stage pre-training applied to MP4 (VANILLA). More implementation details of pre-training are provided in Appendix D.

**Downstream Datasets.** We evaluate the performance of MP4 on open-domain dialogue summarization datasets from two scenarios (i.e., Online-Chat and Daily-Life), namely SAMSum (Gliwa et al., 2019) and DIALOGSUM (Chen et al., 2021), as well as a customer service dialogue summarization dataset from a specific domain (i.e., Tweet), namely TWEETSUMM (Feigenblat et al., 2021). Table 1 provides the statistics of the downstream datasets. More details are provided in Appendix B.

**Comparison Methods.** We compare MP4 with three types of baselines: extractive models, abstractive models, and previous SOTA models. The following presents the comparison methods.

- **Extractive Models.** Based on heuristic algorithms or graph-based algorithms, including **LONGEST**, **Lead-3**, and **TextRank** (Mihalcea and Tarau, 2004).

- **Abstractive Models.** Based on neural network sequence-to-sequence models, including **PGNet** (See et al., 2017), **FastAbs-RL** (Chen and Bansal, 2018), and **Transformer** (Vaswani et al., 2017).

- **Previous SOTA Models.** State-of-the-art dialogue summarization models based on pre-trained models, including **BART($\mathcal{D}_{\textbf{ALL}}$)** (Feng et al., 2021), **Coref-ATTN** (Liu et al., 2021), **BART-ConFiT** (Tang et al., 2022), **DialSent-PGG** (Jia et al., 2022), **BART-NARR** (Xu et al., 2022), **MV-BART** (Chen and Yang, 2020), **ReWriteSum** (Fang et al., 2022), **BART-SCL** (Geng et al., 2022), **UNILMV2** (Bao et al., 2020), **BART** (Lewis et al., 2020), **LA-BART** (Wang et al., 2022a), and **BART-MT** (Bhattacharjee et al., 2022).

**Evaluation Metrics.** We evaluate the full fine-tuning, zero-shot, and few-shot performance of all models using ROUGE scores[4] (i.e., R-1, -2, and -L), which are standard evaluation metrics.

---

[3] https://huggingface.co/facebook/bart-large

[4] https://pypi.org/project/py-rouge

| Model | R-1 | R-2 | R-L |
|---|---|---|---|
| *Extractive and Abstractive Models* | | | |
| TextRank (Mihalcea and Tarau, 2004) | 29.27 | 8.02 | 28.78 |
| PGNet (See et al., 2017) | 37.27 | 14.42 | 34.36 |
| FastAbs-RL (Chen and Bansal, 2018) | 41.03 | 16.93 | 39.05 |
| Transformer (Vaswani et al., 2017) | 42.37 | 18.44 | 39.27 |
| *Previous SOTA Models* | | | |
| BART($\mathcal{D}_{ALL}$) (Feng et al., 2021) | 53.70 | 28.79 | 50.81 |
| Coref-ATTN (Liu et al., 2021) | 53.93 | 28.58 | 50.39 |
| BART-ConFiT (Tang et al., 2022) | 53.89 | 28.85 | 49.29 |
| DialSent-PGG (Jia et al., 2022) | 53.54 | 28.91 | 50.21 |
| BART-NARR (Xu et al., 2022) | 53.80 | 28.96 | 50.76 |
| MV-BART (Feng et al., 2022) | 54.05 | 28.56 | 50.57 |
| ReWriteSum (Fang et al., 2022) | 54.20 | 27.10 | 50.10 |
| BART-SCL (Geng et al., 2022) | 54.22 | 29.87 | 51.35 |
| *Our Models* | | | |
| BART-large | 53.32 | 28.78 | 50.63 |
| MP4 (VANILLA) | 53.41 | 29.23 | 50.97 |
| MP4 (DAP) | 53.82 | 29.55 | 51.21 |
| MP4 (TOP) | 54.56 | 30.33 | 51.70 |
| MP4 (DAP-TOP) | **54.60** | **30.57** | **51.79** |

Table 2: Full fine-tuning results on SAMSum test set.

## 4.2 Full Fine-Tuning Evaluation

To demonstrate the advantages of our pre-trained model with a large amount of training samples, we train the model using the entire training set for full fine-tuning evaluation.

**Settings.** We provide all the hyper-parameters used for fine-tuning and inference in Appendix E. During the evaluation, for SAMSum, we followed (Liu and Lapata, 2019) by testing with the top-3 best checkpoints on the validation set and reporting the average ROUGE scores. For DIALOGSUM, we followed (Chen et al., 2021) by reporting the average ROUGE scores between the inference output and multiple reference summaries. For TWEET-SUMM, due to limited research and the lack of detailed evaluation procedures in the original paper (Feigenblat et al., 2021), we suggest the following evaluation method: (1) During training, use the reference summary with the highest ROUGE-Avg score (i.e., the average value of R-1, R-2, and R-L) between the original dialogue and multiple reference summaries for training. (2) During testing, calculate the average ROUGE scores between the inference output and multiple reference summaries.

**Results.** Since most previous SOTA models have not been evaluated on a wide range of dialogue summarization datasets, therefore, we present the full fine-tuning performance on the SAMSum, DI-ALOGSUM, and TWEETSUMM test sets in Tables 2, 3, and 4, respectively. Compared to previous SOTA models, our MP4 (DAP-TOP) achieves new

| Model | R-1 | R-2 | R-L |
|---|---|---|---|
| *Extractive and Abstractive Models* | | | |
| LONGEST | 24.10 | 6.20 | 22.70 |
| Lead-3 | 27.50 | 6.80 | 27.30 |
| Transformer (Vaswani et al., 2017) | 35.91 | 8.74 | 33.50 |
| *Previous SOTA Models* | | | |
| UNILMv2-base (Bao et al., 2020) | 47.04 | 21.13 | 45.04 |
| BART-large (Lewis et al., 2020) | 47.28 | 21.18 | 44.83 |
| BART-NARR (Xu et al., 2022) | 47.52 | 20.82 | 45.10 |
| LA-BART (Wang et al., 2022a) | 47.28 | 21.09 | 45.11 |
| BART-MT (Bhattacharjee et al., 2022) | 47.26 | 21.18 | 45.17 |
| *Our Models* | | | |
| BART-large | 46.68 | 20.96 | 44.68 |
| MP4 (VANILLA) | 46.87 | 21.21 | 44.87 |
| MP4 (DAP) | 47.01 | 21.37 | 44.94 |
| MP4 (TOP) | 47.74 | **21.84** | 45.77 |
| MP4 (DAP-TOP) | **48.01** | 21.72 | **45.92** |

Table 3: Full fine-tuning results on DIALOGSUM test set. We report the average of multiple-reference results.

| Model | R-1 | R-2 | R-L |
|---|---|---|---|
| DistilBART (Feigenblat et al., 2021)* | 37.94 | 19.26 | 33.51 |
| BART-large (*ours*) | 45.85 | 22.14 | 44.77 |
| MP4 (VANILLA) | 45.56 | 22.31 | 44.84 |
| MP4 (DAP) | 46.61 | 22.82 | 45.08 |
| MP4 (TOP) | 46.84 | **23.63** | 45.74 |
| MP4 (DAP-TOP) | **46.93** | 23.60 | **45.82** |

Table 4: Full fine-tuning results on TWEETSUMM test set. * denotes results obtained from (Feigenblat et al., 2021).

state-of-the-art results on all metrics across three downstream datasets from different scenarios and domains, demonstrating significant performance improvements. Specifically, on SAMSum and DI-ALOGSUM, MP4 (DAP-TOP) surpasses the previous SOTA models by improving the R-2 score by 0.70 and 0.54 (29.87→30.57 and 21.18→21.72), respectively. The improvement on TWEETSUMM reaches 1.46 (22.14→23.60). This indicates that multi-stage pre-training can assist the model better adapt to downstream datasets from a wide range of scenarios and domains. Additionally, compared to BART-large, MP4 (VANILLA) demonstrates stronger performance on most metrics, proving the effectiveness of introducing dialogue modeling. Furthermore, compared to MP4 (VANILLA), the results of MP4 (DAP) indicate that domain-aware pre-training can enhance the adaptability of model to dialogues in multiple scenarios and domains. Moreover, MP4 (TOP) achieves significant performance improvements, highlighting the importance of equipping the model with the ability to summarize multi-scenario multi-domain dialogues during the pre-training stage.

| Model | SAMSum | | DialogSum | | TweetSumm | |
|---|---|---|---|---|---|---|
| | *zero-shot* | *few-shot* (10) | *zero-shot* | *few-shot* (10) | *zero-shot* | *few-shot* (10) |
| BART-large (*ours*) | 27.94/8.56/26.28 | 39.47/16.51/38.76 | 25.50/6.10/25.32 | 36.75/12.98/36.04 | 29.42/10.68/28.18 | 43.14/19.46/41.75 |
| MP4 (DAP) | 32.48/9.80/31.03 | 41.75/17.34/40.29 | 27.80/6.61/27.42 | 36.89/12.88/36.03 | 31.76/11.30/31.80 | 43.98/20.08/42.31 |
| MP4 (DAP-TOP) | **42.41/17.15/39.69** | **48.73/23.63/45.96** | **38.76/14.59/37.18** | **40.18/16.22/39.11** | **38.92/13.16/35.11** | **45.09/20.20/43.03** |

Table 5: R-1/R-2/R-L results in zero-shot and few-shot settings. For zero-shot setting, we report the results at the optimal summary length limits. For few-shot setting, we report the average results from 5 random runs on 10 training samples (all models share the same seed set).

## 4.3 Zero- and Few-Shot Evaluation

Many existing studies that apply pre-trained models to dialogue summarization require a large amount of fine-tuning data, which is often impractical in new scenarios or domains. In contrast, we expect our model to quickly adapt to new scenarios or domains without the need for a large amount of fine-tuning data. To validate this hypothesis, we conduct evaluations in zero-shot (no training samples) and few-shot (10 training samples) settings. Obtaining such a small number of samples is feasible in practice for new scenarios or domains.

**Settings.** We compare the performance of BART-large (Lewis et al., 2020), MP4 (DAP), and MP4 (DAP-TOP) in zero-shot and few-shot settings. In Appendix F, we provide all the hyper-parameters used. Specifically, for zero-shot evaluation, since the models have not been trained on downstream datasets, we report the results of using the optimal summary length limits during inference. For few-shot evaluation, we randomly sample 10 training samples for training. Additionally, to ensure that the results are not affected by sampling variability, we conduct the same experiment five times with different random seeds (shared among all models) and report the average results.

**Results.** The results presented in Table 5 indicate that our pre-trained model achieves significant improvements compared to BART-large. Specifically, for zero-shot results, MP4 (DAP-TOP) increases the R-1 score by 14.47 (27.94→42.41), 13.26 (25.50→38.76), and 9.50 (29.42→38.92) on SAMSum, DialogSum, and TweetSumm, respectively. Moreover, the zero-shot performance of MP4 (DAP-TOP) surpassed the few-shot performance of BART-large on multiple datasets, demonstrating its powerful zero-shot capability. Additionally, the few-shot results also highlight the advantages of MP4 (DAP-TOP). indicating that our pre-trained model converges faster than other models even with only 10 training samples.

| Model | R-1 | R-2 | R-L |
|---|---|---|---|
| MP4 (VANILLA) | 53.41 | 29.23 | 50.97 |
| w/o speaker embeddings | 53.29 | 29.02 | 50.85 |
| MP4 (DAP) | 53.82 | 29.55 | 51.21 |
| w/o token-level tasks | 53.74 | 29.25 | 51.05 |
| w/o utterance infilling | 53.75 | 29.31 | 51.06 |
| w/o utterance permutation | 53.69 | 29.38 | 51.08 |
| w/o utterance masking | 53.53 | 29.30 | 50.94 |
| MP4 (TOP) | 54.56 | 30.33 | 51.70 |
| w/o CSDS pre-training corpus | 54.47 | 30.12 | 51.69 |
| MP4 (DAP-TOP) | **54.60** | **30.57** | **51.79** |

Table 6: Ablation study on SAMSum in full fine-tuning setting. The token-level tasks refer to Token Masking and Token Deletion.

## 4.4 Ablation Study

To further validate the contributions of the fine-grained components in our pre-trained models, we conduct an ablation study on SAMSum in full fine-tuning setting. Table 6 shows the evaluation results.

**Speaker Embeddings.** As the results show, incorporating additional speaker embeddings in dialogue modeling can capture the underlying role interactions during the dialogue process and improve the performance of dialogue summarization.

**Domain-Aware Pre-training Objectives.** As the results show, each domain-aware pre-training objective brings performance improvements. It is worth noting that the utterance masking task has the greatest impact on performance, indicating that completing principal Gap-utterances during dialogue reconstruction is crucial for dialogue summarization.

**Impact of CSDS Pre-training Corpus on ODDS.** We remove the customer service "*dialogue-summary*" parallel data from Lcm³Ds to investigate the impact of this portion of data on the performance of open-domain dialogue summarization. The results show that the model trained without this data exhibits a slight decrease in performance. One possible reason is that a few dialogues in SAMSum also involve customer service topics.

| Model | Flu. | Conci. | Info. | Comp. |
|---|---|---|---|---|
| ChatGPT (*zero-shot*) | **2.24** | 3.82 | 3.55 | **1.37** |
| BART-large (*ours*) | 2.77 | 2.19 | 2.48 | 3.13 |
| MP4 (DAP-TOP) | 2.53 | **2.02** | **2.14** | 3.08 |
| Ground Truth | 2.46 | 1.97 | 1.83 | 2.42 |

Table 7: Human evaluation on SAMSum test set.

## 5 Human Evaluation

We conduct human evaluation to further evaluate the performance of our pre-trained model and strong baselines under various paradigms, as well as the Ground Truth (i.e., MP4 (DAP-TOP), ChatGPT, BART-large, Ground Truth). Specifically, we randomly select 50 samples from the test set of SAMSum. Then, we invite 3 participants to rank four candidate summaries according to four metrics: *fluency* (Flu.), *conciseness* (Conci.), *informativeness* (Info.), and *comprehensiveness* (Comp.). The top-ranking indicates the best performance on that metric.

Table 7 shows the results of human evaluation (lower average rank is better). Our pre-trained model outperforms BART-large in all metrics but falls behind the Ground Truth. Specifically, ChatGPT achieves the first rank in *fluency* and *comprehensiveness* for the summaries generated in zero-shot setting, surpassing the Ground Truth. However, it exhibits the weakest performance in *conciseness* and *informativeness*. The main reason for this is that ChatGPT tends to generate longer summaries that describe various aspects of the dialogue, including both important and minor details. Moreover, the longer summaries also contribute to an improved overall impression to some extent.

## 6 Related Work

**Dialogue Corpora.** In general, dialogue data can be obtained from two main sources. One is massive-scale dialogue corpora crawled from web platforms such as Reddit (Baumgartner et al., 2020; Henderson et al., 2019) and Twitter (Ritter et al., 2010), which are commonly used for pre-training open-domain chatbots (Zhang et al., 2020b; Adiwardana et al., 2020; Roller et al., 2021; Chen et al., 2022; Henderson et al., 2020). Another source is a collection of open-source dialogue datasets designed for specific tasks, including open-domain dialogue systems (Li et al., 2017; Zhou et al., 2018; Dinan et al., 2018; Smith et al., 2020; Zhang et al., 2018; Gopalakrishnan et al., 2019; Komeili et al., 2022;

Dinan et al., 2019), task-oriented dialogue systems (Lee et al., 2019; Rastogi et al., 2020; Byrne et al., 2019; Zang et al., 2020; Li et al., 2018; Feng et al., 2020), dialogue comprehension (Sun et al., 2019; Yang and Choi, 2019; Yu et al., 2020; Cui et al., 2020; Nie et al., 2021; Rashkin et al., 2019), and dialogue summarization (Gliwa et al., 2019; Chen et al., 2021; Feigenblat et al., 2021).

**PTMs for Dialogue Summarization.** Recently, general-purpose pre-trained models have achieved significant success in dialogue summarization tasks (Lewis et al., 2020; Raffel et al., 2020; Bao et al., 2020; Beltagy et al., 2020). Furthermore, several task-specific pre-trained models (Zhang et al., 2020a; Zhong et al., 2022) have further improved dialogue summarization. Moreover, existing state-of-the-art dialogue summarization models typically leverage pre-trained models and model the characteristics of dialogues to achieve better results, including modeling dialogue interactions (Lin et al., 2022; Tang et al., 2022), incorporating extra information (Wang et al., 2022c; Kim et al., 2022), and dialogue rewriting (Xu et al., 2022; Fang et al., 2022). Although these models are effective, they often have complex model structures that are difficult to apply within the current pre-training paradigm.

**Large Language Models.** More recently, LLMs have attracted widespread attention due to their remarkable performance in various knowledge-intensive NLP tasks (Zeng et al., 2022; Ouyang et al., 2022; Thoppilan et al., 2022; Scao et al., 2022). Through large-scale pre-training on massive text corpora (Brown et al., 2020; Wei et al., 2022b), LLMs possess powerful foundational capabilities. Instruction tuning (Raffel et al., 2020; Wei et al., 2022a; Chung et al., 2022) helps LLMs in understanding natural language task descriptions. while Reinforcement Learning with Human Feedback (RLHF) (Stiennon et al., 2020; Bai et al., 2022) aligns generated text with human preferences.

## 7 Conclusion

In this study, we propose MP4, a multi-stage pre-trained model for multi-scenario multi-domain dialogue summarization. To conduct the pre-training, we construct a large-scale ChatGPT-annotated multi-scenario multi-domain multi-turn dialogue summarization corpus called LCM$^3$DS. Extensive experimental results demonstrate that MP4 exhibits remarkable dialogue summarization capabilities.

## Limitations

Although we have demonstrated the powerful performance of MP4 in multi-scenario multi-domain dialogue summarization, there are still some limitations that provide directions for future work: (1) Due to limitations in computational resources, we did not consider long dialogues when constructing LCM³DS. Therefore, MP4 may be more suitable for short dialogue summarization. (2) MP4 is initialized with BART-large, which has only 0.4 billion parameters. In future work, we will consider using larger base models.

## Ethics Statement

All dialogue data used in this study are sourced from previously published works, and there are no copyright restrictions on their academic use, allowing free online access. Since we used ChatGPT for data annotation, the LCM³DS corpus we constructed is intended solely for academic research purposes. Additionally, MP4 is initialized with the weights from BART-large. Therefore, MP4 may exhibit biases and harmful behaviors commonly observed in language models.

## Acknowledgement

This work received support from both the National Natural Science Foundation of China (Grant Nos. 62276017, U1636211, 61672081) and the Fund of the State Key Laboratory of Software Development Environment (Grant No. SKLSDE-2021ZX-18). We extend our sincere appreciation to Yu Zhou and Rongping Chang for their valuable contributions to the revision of this paper.

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

# A Prompts for Dialogue Summarization

We utilize ChatGPT as an unsupervised summarizer to annotate dialogues in the SAMSum (Gliwa et al., 2019) test set using three different prompt formats in zero-shot setting:

- **Preceding Prompt.** Insert the prompt "*Summarize the following dialogue into a short summary:*" before the dialogue.

- **InstructGPT Prompt.** Following Instruct-GPT (Ouyang et al., 2022) insert the prompt "*Tl;dr:*" after the dialogue.

- **Subsequent Prompt.** Insert the prompt "*Summarize the above dialogue into a short summary:*" after the dialogue.

| Prompt | Rouge-1 | Rouge-2 | Rouge-L |
|---|---|---|---|
| Preceding | 37.90 | 15.19 | 35.89 |
| InstructGPT | **42.17** | **16.84** | **39.26** |
| Subsequent | 40.08 | 15.41 | 37.22 |

Table 8: Comparison of dialogue summarization performance for different prompts on SAMSum test set, which contains 819 dialogues.

Table 8 shows the evaluation results of three different prompts. The **InstructGPT Prompt** achieves the best performance.

# B Details of Downstream Datasets

**SAMSum.** A natural messenger dialogue summarization dataset containing dialogues created and written down by linguists fluent in English. The dataset includes various topics such as chit-chats, gossiping about friends, arranging meetings, discussing politics, consulting university assignments with colleagues, etc. The sizes of the training, validation, and test sets are 14,731, 818, and 819 respectively.

**DIALOGSUM.** A large-scale dataset for dialogue summarization, which includes face-to-face spoken dialogues covering a wide range of daily-life topics, including schooling, work, medication, shopping, leisure, and travel, etc. Most dialogues take place between friends, colleagues, and between service providers and customers. The sizes of the training, validation, and test sets are 12,460, 500, and 500 respectively.

**TWEETSUMM.** A customer service dialogue summarization dataset, consisting of reconstructed dialogues extracted from the Kaggle Customer Support On Twitter dataset. The Kaggle dataset is a large-scale dataset based on dialogues between consumers and customer support agents on Twitter.com. It covers a wide range of topics and services provided by various companies, including airlines, retail, gaming, music, etc. The sizes of the training, validation, and test sets are 869, 108, and 110 respectively.

# C Example of Data Augmentation

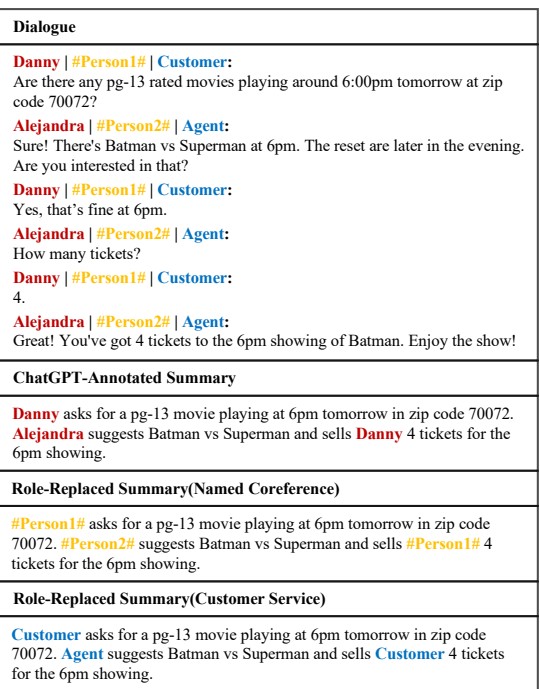

Figure 3: An example of role-replaced dialogue and summary. *Red* indicates a role group added through role adding, *yellow* indicates named coreference, and *blue* indicates roles in customer service.

# D Implementation Details of Pre-training

The following are the hyper-parameters used in domain-aware pre-training.

$-gpus$ 8
$-steps$ 5000
$-batch\_size$ 16
$-lr$ $3e-05$
$-warmup\_steps$ 500
$-label\_smoothing$ 0.1
$-optimizer$ Adam

The following are the hyper-parameters used in task-oriented pre-training.

$-gpus$ 8
$-steps$ 10000
$-batch\_size$ 16
$-lr$ $3e-05$
$-warmup\_steps$ 1000
$-label\_smoothing$ 0.1
$-optimizer$ $Adam$

## E  Details of Fine-Tuning and Inference

The following are the hyper-parameters used in full fine-tuning setting on SAMSum.

$-gpus$ 4
$-steps$ 1150
$-batch\_size$ 16
$-lr$ $3e-05$
$-warmup\_steps$ 100
$-label\_smoothing$ 0.1
$-optimizer$ $Adam$

The following are the hyper-parameters used in full fine-tuning setting on DIALOGSUM.

$-gpus$ 4
$-steps$ 1000
$-batch\_size$ 16
$-lr$ $3e-05$
$-warmup\_steps$ 100
$-label\_smoothing$ 0.1
$-optimizer$ $Adam$

The following are the hyper-parameters used in full fine-tuning setting on TWEETSUMM.

$-gpus$ 4
$-steps$ 98
$-batch\_size$ 16
$-lr$ $3e-05$
$-warmup\_steps$ 10
$-label\_smoothing$ 0.1
$-optimizer$ $Adam$

All models maintain consistent hyper-parameters across all datasets during inference.

$-gpus$ 1
$-batch\_size$ 32
$-use\_cache$ $true$
$-max\_length$ 100
$-min\_length$ 5
$-beam\_size$ 5
$-length\_penalty$ 1
$-no\_repeat\_ngram\_size$ 0
$-early\_stopping$ $false$

## F  Zero- and Few-Shot Evaluation Details

For zero-shot evaluation, the optimal summary length limit hyper-parameter $max\_length$ for SAMSum, DIALOGSUM, and TWEETSUMM is 60, 40, and 80 respectively. Moreover, other hyper-parameters used during inference remain consistent with Appendix E. For few-shot evaluation (with 10 training samples), we provide the hyper-parameter settings used during training below, while the hyper-parameters used during inference are consistent with Appendix E.

$-gpus$ 1
$-steps$ 20
$-batch\_size$ 10
$-lr$ $3e-05$
$-warmup\_steps$ 0
$-label\_smoothing$ 0.1
$-optimizer$ $Adam$
$-seeds$ 3442|3443|3444|3445|3446

## G  Data Statistics for Dialogue Datasets

Please refer to Table 9.

## H  Examples of Generated Summaries

Please refer to Figure 4.

| Dataset | #Dialogue | #Utterance | #Turns/dial. | #Tokens/dial. | #Tokens/turn | #Tokens/summ. | Sce./#Dom. |
|---|---|---|---|---|---|---|---|
| *Open-Domain Dialogue* | | | | | | | |
| Empathetic Dialogues (Rashkin et al., 2019) | 22,878 | 98,759 | 4.3 | 86.5 | 20.1 | 34.2 | Empathetic |
| Decode (Nie et al., 2021) | 22,119 | 181,580 | 8.2 | 157.6 | 19.2 | 47.3 | Multiple |
| Wizard of Wikipedia (Dinan et al., 2019) | 19,287 | 174,624 | 9.1 | 213.7 | 23.5 | 55.8 | Wizard |
| Wizard of Internet (Komeili et al., 2022) | 9,025 | 91,042 | 10.1 | 225.0 | 22.3 | 55.6 | Wizard |
| Topical-Chat (Gopalakrishnan et al., 2019) | 6,524 | 140,544 | 21.5 | 507.7 | 23.6 | 61.2 | Topical |
| Persona-Chat (Zhang et al., 2018) | 5,648 | 84,366 | 14.9 | 239.9 | 16.1 | 55.0 | Persona |
| Blended Skill Talk (Smith et al., 2020) | 4,662 | 62,581 | 13.4 | 268.1 | 20.0 | 53.8 | Multiple |
| ConvAI2 (Dinan et al., 2018) | 4,554 | 67,532 | 14.8 | 237.8 | 16.1 | 53.9 | Persona |
| MuTual (Cui et al., 2020) | 2,719 | 14,134 | 5.2 | 98.7 | 19.0 | 38.1 | Daily |
| CMU DoG (Zhou et al., 2018) | 2,647 | 58,406 | 22.1 | 422.4 | 19.1 | 67.9 | Movie |
| DailyDialog (Li et al., 2017) | 2,471 | 13,891 | 5.6 | 79.0 | 14.1 | 31.4 | Daily |
| DialogRE (Yu et al., 2020) | 1,578 | 20,724 | 13.1 | 249.6 | 19.1 | 49.2 | Chit-Chat |
| FriendsQA (Yang and Choi, 2019) | 969 | 16,415 | 16.9 | 307.1 | 18.2 | 50.1 | Chit-Chat |
| DREAM (Sun et al., 2019) | 345 | 1,706 | 4.9 | 74.3 | 15.2 | 32.1 | Daily |
| *Customer Service Dialogue* | | | | | | | |
| MetaLWOz (Lee et al., 2019) | 37,860 | 431,580 | 11.4 | 146.0 | 12.8 | 45.2 | 47 |
| Schema (Rastogi et al., 2020) | 22,559 | 453,320 | 20.1 | 322.7 | 16.1 | 65.1 | 17 |
| Taskmaster2 (Byrne et al., 2019) | 17,083 | 286,930 | 16.8 | 264.7 | 15.8 | 59.8 | 7 |
| MultiWOZ 2.2 (Zang et al., 2020) | 10,330 | 141,162 | 13.7 | 280.4 | 20.5 | 62.6 | 8 |
| MSR-E2E (Li et al., 2018) | 8,769 | 68,406 | 7.8 | 150.0 | 19.2 | 46.9 | 3 |
| doc2dial (Feng et al., 2020) | 4,741 | 56,221 | 11.9 | 260.7 | 21.9 | 76.9 | 4 |

Table 9: Data statistics for preprocessed, cleaned and annotated dialogue datasets of LCM³DS. # denotes average value. *Dial.* represents dialogue. *Summ.* represents ChatGPT-annotated summary. *Sce.* represents scenario of open-domain dialogue. *Dom.* represents domains of customer service dialogue.

---

**Dialogue**

**Peyton:** I have been asking you to bring that video game for me.
**Cameron:** Honey, I am not having enough time to come home.
**Peyton:** When would you come home?
**Cameron:** I will have to stay out of town for another week i guess.
**Peyton:** Cant you just deliver that game through the courier? :P.
**Cameron:** Dont be mean :/
**Peyton:** Get the job done and come to home then. ASAP :P.

**ChatGPT (*zero-shot*)**

Peyton asks Cameron to bring a video game, but Cameron is out of town and unable to come home. Peyton suggests delivering the game through courier, but Cameron refuses. Cameron will be out of town for another week and Peyton tells him to get the job done and come home ASAP.

**BART-large**

Cameron will have to stay out of town for another week. He will have to deliver the video game to Peyton.

**MP4 (DAP-TOP)**

Peyton wants Cameron to bring her a video game. Cameron will be out of town for another week.

**Ground Truth**

Peyton is expecting Cameron to bring the video game. Cameron will probably be out for another week.

---

**Dialogue**

**Freddie:** What are you watching on Netflix? I've just finished Mad Men.
**Kelly:** I've just finished The Crown.
**Freddie:** There should be new episodes coming soon.
**Kelly:** OMG! I can't wait! :)
**Freddie:** Try Outlander or The Tudors in the meantime.
**Jim:** I'm watching I, Zombie.
**Bob:** Try Grimm or Sabrina. They're both great!
**Kelly:** They are all horrible and disgusting!
**Bob:** I think they are funny. They're just TV shows don't take them too seriously. X.
**Greg:** I'm watching House of Cards though must admit the new series is boring.
**Mike:** I agree. Not the same without Kevin Spacey! You should try The West Wing if you enjoy political dramas.
**Greg:** Netflix original Bodyguard is supposed to be good from what I've heard.
**Mary:** I'm a bit ashamed to confess i enjoy stuff like Pretty Little Liars or Gossip Girl ;)
**Nancy:** Don't be ashamed i enjoy them too! And my favourite one is Desperate Housewives!!!
**Mary:** I know! I watched it twice!
**Alan:** I'm catching up with Friends.
**Nancy:** OMG! I used to love Friends!

**ChatGPT (*zero-shot*)**

A group of people discuss their favorite shows on Netflix, recommend new ones to each other, and share their opinions on different genres. They mention shows like Mad Men, The Crown, Outlander, The Tudors, I, Zombie, Grimm, Sabrina, House of Cards, The West Wing, Bodyguard, Pretty Little Liars, Gossip Girl, Desperate Housewives, and Friends.

**BART-large**

Freddie has just finished watching Mad Men. Kelly has just finished The Crown. Jim is watching I, Zombie. Greg is watching House of Cards. Alan is catching up with Friends. Nancy's favourite series is Desperate Housewives.

**MP4 (DAP-TOP)**

Freddie, Kelly, Jim, Bob, Greg, Mary and Alan are discussing what they are watching on Netflix.

**Ground Truth**

Freddie, Kelly, Jim, Greg, Bob, Mike, Mary, Alan and Nancy are watching different shows on Netflix.

Figure 4: Examples of generated summaries.