# OpenReview forum: "Multi-Stage Pre-training Enhanced by ChatGPT for Multi-Scenario Multi-Domain Dialogue Summarization"
_EMNLP/2023/Conference — EMNLP 2023 Findings_

### Official Review · Reviewer_BDoc · 2023-08-04

**Soundness:** 4

**Excitement:**

3: Ambivalent: It has merits (e.g., it reports state-of-the-art results, the idea is nice), but there are key weaknesses (e.g., it describes incremental work), and it can significantly benefit from another round of revision. However, I won't object to accepting it if my co-reviewers champion it.

**Paper Topic And Main Contributions:**

The paper focuses on multi-scenario multi-domain dialogue summarization. It proposes a large-scale dataset, a multi-stage pre-training strategy. The experimental results shows the effectiveness of the proposed data and the pre-training method.

**Reasons To Accept:**

1. This paper proposes a large-scale multi-scenario multi-domain dialogue data annotated by ChatGPT, indicating a successful application of LLM to enhance small models.
2. It provides a valuable datasets for pre-training and evaluating dialogue summarization models.
3. The paper conducts extensive experiments and analyses to prove the effectiveness of the proposed method. The experimental results show the method achieves SOTA results. The zero-shot and few-shot experiments suggest the capability of multiple scenarios and domains adaptation.

**Reasons To Reject:**

1. This work relies on ChatGPT for data annotation, which may introduce biases or errors. Human annotation could be more accurate.
2. The pre-training strategies are similar to BART, lacking novelty.

**Reproducibility:**

4: Could mostly reproduce the results, but there may be some variation because of sample variance or minor variations in their interpretation of the protocol or method.

**Reviewer Confidence:**

4: Quite sure. I tried to check the important points carefully. It's unlikely, though conceivable, that I missed something that should affect my ratings.

---

> ### Author Rebuttal · Authors · 2023-08-26
>
> Thanks for your thoughtful and valuable comments. We will explain your concerns and questions as follows:
>
> Q1: This work relies on ChatGPT for data annotation, which may introduce biases or errors. Human annotation could be more accurate.
>
> A1: We agree that human-annotated summaries might be more accurate, as they are often considered the gold standard. However, we chose to use ChatGPT for data annotation for several reasons:
>
> 1. One of our primary goals is to investigate whether by annotating large-scale multi-scenario multi-domain dialogue summarization pre-training corpus with ChatGPT, we can significantly improve the transfer performance of small-scale pre-trained models on multi-scenario multi-domain downstream datasets.
>
> 2. Human annotations, especially on a large scale, are typically resource-intensive and time-consuming. Leveraging LLMs can considerably reduce costs. This not only makes the deployment of enhanced small-scale models in practical projects more feasible but also aids in rapid iterations and optimizations.
>
> 3. While human annotation might be seen as more accurate, it is susceptible to inconsistencies and subjectivities across different annotators. LLMs, with their consistent annotation approach, can alleviate such variability. Although LLMs might introduce a small amount of potential biases or errors, these can be further reduced as our pre-trained model undergoes further fine-tuning on human-annotated downstream datasets.
>
> We deeply understand the reviewer's concerns regarding the choice of data annotation method. In this study, we balanced transfer efficiency against accuracy and believe that ChatGPT annotation is an effective solution to achieve our goals. In the future, we aim to explore the integration of human annotations, combining the efficiency and consistency of LLM annotations with the accuracy of human annotations to further optimize model performance and robustness.
>
> Q2: The pre-training strategies are similar to BART, lacking novelty.
>
> A2: We appreciate your insights regarding the similarities between our domain-aware pre-training and BART. Although our method indeed draws inspiration from BART, we would like to clarify the fundamental reasons for our choices and several distinctions:
>
> 1. One of the main focuses of our study is to bridge the gap between pre-training and fine-tuning objectives in dialogue summarization tasks using a multi-stage pre-training strategy. During the domain-aware pre-training (DAP) stage, we aim for the model to focus on understanding dialogues across various scenarios and domains, while avoiding learning knowledge specific to particular tasks. In this way, the model can still adapt to other dialogue-related tasks, such as dialogue sentiment analysis, response generation, and dialogue information extraction. Hence, we have adopted more general pre-training tasks similar to BART.
>
> 2. We adjusted the masking ratio for the Token Masking and Token Deletion tasks, increasing it from 15% to 20%, because the information density of dialogues is usually lower than that of pure texts. Additionally, our Utterance Permutation task differs from previous related works, as we only shuffled the order of utterances, hoping that the model can accurately align utterances with roles. Moreover, we designed a new Utterance Masking task, where the model needs to reconstruct dialogues with masked Gap-utterances.
>
> Thank you again for your constructive comments. We will further emphasize these differences in the revised version.

---

### Official Review · Reviewer_SZZj · 2023-08-04

**Soundness:** 3

**Excitement:**

2: Mediocre: This paper makes marginal contributions (vs non-contemporaneous work), so I would rather not see it in the conference.

**Paper Topic And Main Contributions:**

This paper focused on the dialogue summarization problem. It first collected 20 dialogue datasets and utilized ChatGPT to annotate the datasets automatically. Then it pre-trained and fine-tuned BART by multi-stage pre-training strategy including domain-aware pre-training, task-oriented pre-training and fine-tuning. Experiments on three public datasets (SAMSum, DIALOGSUM, and TWEETSUMM) showed the effectiveness of the proposed approach.

**Questions For The Authors:**

1. What's the main contribution of this work? The distilled dataset?

**Reasons To Accept:**

1. The experimental results look competitive.

**Reasons To Reject:**

1. Lack of novelty. Since the domain-adaptive post-training has been proposed before, this main contribution of this work is like weak supervision with ChatGPT or knowledge distillation. For the model level, there is not too much innovation.
2. In Section 2, it mentioned that it added roles for the original dialogue datasets. I would wonder if the operation is also the same for those baselines, especially for the previous works illustrated in the Table 2~4.


**Reproducibility:**

3: Could reproduce the results with some difficulty. The settings of parameters are underspecified or subjectively determined; the training/evaluation data are not widely available.

**Reviewer Confidence:**

4: Quite sure. I tried to check the important points carefully. It's unlikely, though conceivable, that I missed something that should affect my ratings.

---

> ### Author Rebuttal · Authors · 2023-08-27
>
> Thanks for your thoughtful and valuable comments. We will explain your concerns and questions as follows:
>
> Q1: What's the main contribution of this work? The distilled dataset?
>
> A1: The main contribution of this work is multifaceted:
>
>   1. Multi-Stage Pre-trained Model (MP4): This model provides both the academic and industrial communities with a pre-trained model that can be rapidly transferred to downstream datasets across various scenarios and domains. On one hand, compared to other dialogue summarization models, it demonstrates outstanding performance and transfer efficiency and integrates both open-domain dialogue summarization and customer service dialogue summarization. On the other hand, in comparison to LLMs, it has only 0.4 billion parameters, significantly reducing computational resource overhead, and also supports further fine-tuning in an open-source manner.
>
> 2. High-Quality Large-Scale Dialogue Summarization Corpus (LCM3DS): This corpus encompasses multi-turn dialogues across a wide range of scenarios and domains. To the best of our knowledge, this is currently the largest, open-sourced, uniformly formatted dialogue collection with the most extensive coverage of scenarios and domains. It can promote the development of various dialogue-related tasks (e.g., dialogue selection, response generation, dialogue comprehension, etc.). Additionally, we have annotated the corresponding summaries using ChatGPT, providing invaluable labels for the field of dialogue summarization. This is reflected not only in pre-training but also in potential future evaluations of dialogue summarization.
>
> 3. Demonstrated the feasibility of enhancing small pre-trained models through LLMs: We believe the advantage of LLMs lies in their powerful inference capability suitable for various downstream tasks; for handling specific tasks, small-scale models are already sufficiently robust. Thus, we sought to endow the small-scale pre-trained models with the knowledge of LLMs in specific tasks. Extensive experimental results validated the feasibility of this approach.
>
> Q2: Lack of novelty. Since the domain-adaptive post-training has been proposed before, this main contribution of this work is like weak supervision with ChatGPT or knowledge distillation. For the model level, there is not too much innovation.
>
> A2: We understand the reviewer's concerns regarding novelty, but we would like to clarify our primary motivations and several distinctions:
>
> 1. We agree that one of the main contributions is to demonstrate the feasibility of enhancing small pre-trained models through LLMs.
>
> 2. One of the main focuses of our study is to bridge the gap between pre-training and fine-tuning objectives in dialogue summarization tasks using a multi-stage pre-training strategy. We didn't dedicate much attention to innovations in model architecture, but rather naturally incorporated speaker embeddings for better dialogue representation.
>
> 3. Regarding domain-adaptive post-training, our goal at this stage is to get the model to focus on understanding dialogues across different scenarios and domains. We adjusted the masking ratio for the Token Masking and Token Deletion tasks, increasing it from 15% to 20%, because the information density of dialogues is usually lower than that of pure texts. Additionally, our Utterance Permutation task differs from previous related works, as we only shuffled the order of utterances, hoping that the model can accurately align utterances with roles. Moreover, we designed a new Utterance Masking task, where the model needs to reconstruct dialogues with masked Gap-utterances.
>
> Q3: In Section 2, it mentioned that it added roles for the original dialogue datasets. I would wonder if the operation is also the same for those baselines, especially for the previous works illustrated in the Table 2~4.
>
> A3: Yes, all baseline models represent each turn of the input dialogue with the role followed by the utterance, for example: "Role: Utterance." or "Role<EOR>Utterance<EOU>". Our MP4 model adopted the second scheme to concatenate the dialogue.
>
> Thank you for your comprehensive comments. We will make the necessary updates in the revised version to further clarify our contributions and address the concerns.

---

### Official Review · Reviewer_k7jr · 2023-08-12

**Soundness:** 3

**Excitement:**

3: Ambivalent: It has merits (e.g., it reports state-of-the-art results, the idea is nice), but there are key weaknesses (e.g., it describes incremental work), and it can significantly benefit from another round of revision. However, I won't object to accepting it if my co-reviewers champion it.

**Paper Topic And Main Contributions:**

This paper introduces a novel approach for multi-scenario multi-domain dialogue summarization. It proposes the MP4 pre-trained model, which employs a multi-stage pre-training strategy to bridge the gap between pre-training and fine-tuning objectives. The model's effectiveness is demonstrated on diverse datasets, showcasing significant performance improvements over existing methods in various settings. Additionally, the paper contributes the LCM3DS corpus, a resource containing multi-scenario multi-domain dialogues and their annotated summaries, enhancing the availability of high-quality training data for this task.

**Reasons To Accept:**

This paper presents a significant contribution to the field of dialogue summarization. The proposed MP4 pre-trained model, accompanied by the LCM3DS corpus, addresses the challenge of adapting dialogue summarization across diverse scenarios and domains. The novel multi-stage pre-training strategy and integration of speaker embeddings are notable strengths, leading to impressive performance improvements.

**Reasons To Reject:**


The paper introduces novel strategies for multi-scenario multi-domain dialogue summarization, but it's important to acknowledge potential drawbacks primarily centered around the introduced complexities. The incorporation of a multi-stage pre-training strategy and speaker embeddings adds intricate layers to the model architecture. This increased complexity could lead to challenges in implementation, such as longer development cycles, higher resource requirements, and potential difficulties in reproducing results.

**Reproducibility:**

3: Could reproduce the results with some difficulty. The settings of parameters are underspecified or subjectively determined; the training/evaluation data are not widely available.

**Reviewer Confidence:**

2: Willing to defend my evaluation, but it is fairly likely that I missed some details, didn't understand some central points, or can't be sure about the novelty of the work.

---

> ### Author Rebuttal · Authors · 2023-08-26
>
> Thanks for your thoughtful and valuable comments. We will explain your concerns and questions as follows:
>
> Q1: The paper introduces novel strategies for multi-scenario multi-domain dialogue summarization, but it's important to acknowledge potential drawbacks primarily centered around the introduced complexities. The incorporation of a multi-stage pre-training strategy and speaker embeddings adds intricate layers to the model architecture. This increased complexity could lead to challenges in implementation, such as longer development cycles, higher resource requirements, and potential difficulties in reproducing results.
>
> A1: We appreciate your concern regarding the complexities introduced by our multi-stage pre-training strategy and speaker embeddings. Here is our response to this:
>
> Firstly, although the model indeed incorporates additional layers and complexities, these were specifically introduced to address the intricacies associated with multi-scenario multi-domain dialogue summarization. The primary goal of this design is to ensure significant and robust performance in dialogues across different scenarios and domains.
>
> Secondly, regarding longer development cycles: We agree that multi-stage pre-training may require additional time. However, it is important to emphasize that once the pre-training is completed, our model can converge on downstream datasets faster than other models and demonstrates a strong performance advantage. Additionally, we will release all the pre-trained models and code. Therefore, for real-world transfer, only minimal time cost and computational resources are required for further fine-tuning, eliminating the need for additional pre-training.
>
> Thirdly, as for the higher resource requirements, we agree that the incorporation of speaker embeddings may lead to a slight increase in inference time complexity. However, compared to dialogue summarization models that incorporate more complex architectures (such as role interactions, dialogue rewriting, extra information, and graph structures), this cost is almost negligible. On the other hand, compared to LLMs with parameter sizes ranging from several billion to hundreds of billions, our model only has 0.4 billion parameters, requiring significantly fewer resources for inference.
>
> Lastly, about the potential difficulties in reproducing results: We understand the concern, especially given the complexities involved. To address this issue, we will release all pre-trained models, datasets, code, and training procedures in the open-source community, ensuring that researchers and practitioners can reproduce our results.

---

### Meta-Review · Area_Chair_dmzo · 2023-09-14

**Recommendation:** 3

**Metareview:**

This paper presents a novel approach to multi-scenario multi-domain dialogue summarization, introducing the MP4 pre-trained model with a multi-stage pre-training strategy to enhance pre-training and fine-tuning alignment. Furthermore, the paper contributes the LCM3DS corpus, featuring multi-scenario multi-domain dialogues and annotated summaries, thereby enriching training data quality for this task. The study encompasses the collection of 20 dialogue datasets, automatic annotation using ChatGPT, and multi-stage pre-training involving domain-aware pre-training, task-oriented pre-training, and fine-tuning of BART. Experimental results on public datasets (SAMSUM, DIALOGSUM, and TWEETSUMM) confirm the effectiveness of this proposed approach, emphasizing the importance of both the data and pre-training methodology. However, as the reviewers pointed out, the novelty contributions were insignificant since several previous papers have explored domain-adaptive post-training.

---

### Decision · Program_Chairs · 2023-10-07

**Decision:**

Accept-Findings

**Comment:**

This paper presents a novel approach to multi-scenario multi-domain dialogue summarization, introducing the MP4 pre-trained model with a multi-stage pre-training strategy to enhance pre-training and fine-tuning alignment. Furthermore, the paper contributes the LCM3DS corpus, featuring multi-scenario multi-domain dialogues and annotated summaries, thereby enriching training data quality for this task. The study encompasses the collection of 20 dialogue datasets, automatic annotation using ChatGPT, and multi-stage pre-training involving domain-aware pre-training, task-oriented pre-training, and fine-tuning of BART. Experimental results on public datasets (SAMSUM, DIALOGSUM, and TWEETSUMM) confirm the effectiveness of this proposed approach, emphasizing the importance of both the data and pre-training methodology. However, as the reviewers pointed out, the novelty contributions were insignificant since several previous papers have explored domain-adaptive post-training.